# New Innovations for the Treatment of Vulvovaginal Atrophy: An Up-to-Date Review

**DOI:** 10.3390/medicina58060770

**Published:** 2022-06-06

**Authors:** Vittoria Benini, Alessandro Ferdinando Ruffolo, Arianna Casiraghi, Rebecca S. Degliuomini, Matteo Frigerio, Andrea Braga, Maurizio Serati, Marco Torella, Massimo Candiani, Stefano Salvatore

**Affiliations:** 1Obstetrics and Gynecology Unit, IRCCS San Raffaele Hospital, Vita-Salute University, 20132 Milan, Italy; benini.vittoria@hsr.it (V.B.); alesruffolo@gmail.com (A.F.R.); casiraghi.arianna@hsr.it (A.C.); degliuomini.rebecca@hsr.it (R.S.D.); candiani.massimo@hsr.it (M.C.); 2ASST Monza, Ospedale San Gerardo, 20900 Monza, Italy; frigerio86@gmail.com; 3Department of Obstetrics and Gynecology, EOC-Beata Vergine Hospital, 6850 Mendrisio, Switzerland; andrea.braga@eoc.ch; 4Department of Obstetrics and Gynecology, Del Ponte Hospital, University of Insubria, 21100 Varese, Italy; maurizio.serati@uninsubria.it; 5Department of Obstetrics and Gynecology, Second Faculty, 80100 Naples, Italy; marcotorella@iol.it

**Keywords:** genitourinary syndrome menopause, vulvovaginal atrophy, vaginal atrophy, vaginal laser, CO_2_ laser, CO_2_ vaginal laser, Erbium YAG laser, er:YAG laser

## Abstract

Vulvovaginal atrophy (VVA) is a chronic progressive disease involving the female genital apparatus and lower urinary tract. This condition is related to hypoestrogenism consequent to menopause onset but is also due to the hormonal decrease after adjuvant therapy for patients affected by breast cancer. Considering the high prevalence of VVA and the expected growth of this condition due to the increase in the average age of the female population, it is easy to understand its significant social impact. VVA causes uncomfortable disorders, such as vaginal dryness, itching, burning, and dyspareunia, and requires constant treatment, on cessation of which symptoms tend to reappear. The currently available therapies include vaginal lubricants and moisturizers, vaginal estrogens and dehydroepiandrosterone (DHEA), systemic hormone therapy, and Ospemifene. Considering, however, that such therapies have some problems that include contraindications, ineffectiveness, and low compliance, finding an innovative, effective, and safe treatment is crucial. The present data suggest great efficacy and safety of a vaginal laser in the treatment of genital symptoms and improvement in sexual function in patients affected by VVA. The beneficial effect tends to be sustained over the long-term, and no serious adverse events have been identified. The aim of this review is to report up-to-date efficacy and safety data of laser energy devices, in particular the microablative fractional carbon dioxide laser and the non-ablative photothermal Erbium-YAG laser.

## 1. Introduction

Vulvovaginal atrophy (VVA) is a common and underreported condition that is associated with decreased estrogenization of the genital tissue [1]. It is a chronic progressive disease involving the female genital apparatus and lower urinary tract and is caused by the decrease in hormonal levels following the onset of menopause or, in some cases, following surgical or pharmacological ovarian failure.

In 2014, during a consensus conference of experts involving the International Society for the Study of Women’s Sexual Health (ISSWSH) and the North American Menopause Society (NAMS) [2], new nomenclature was proposed. From that moment on, the terms vulvovaginal atrophy and atrophic vaginitis were replaced by genitourinary syndrome of menopause (GSM), which was considered to be more accurate and all-encompassing.

Among post-menopausal women, the prevalence of this syndrome is estimated to be at least 50%, but some research describes even 80% of women in menopause complaining of at least one symptom referable to GSM [3]. In addition to that, VVA is reported to be a major factor for quality-of-life impairment in breast cancer patients, additionally being a side effect of its adjuvant therapies [4].

Despite its frequency, physicians’ widespread poor awareness regarding VVA and the tendency to consider the symptoms as part of the normal ageing process [5] lead to a significant underestimation of this syndrome‘s real prevalence. Studies investigating the impact of genital disorders among women with GSM have reported complaints regarding sexual life impairments, negative consequences on relationship and marriage, lower quality of life, and negative impacts on self-esteem and social interactions [6]. Moreover, considering the increase in average age, especially among women, it is easy to see that vaginal atrophy represents a major health problem.

The most common symptom is represented by vaginal dryness (reported by more than half to 83% of post-menopausal women depending on the study) [6,7], followed by dyspareunia (38–59%) and vaginal irritation (37–77%) [5,8]. On the other hand, urinary symptoms include urgency, frequency, urinary incontinence, dysuria, and recurrent urinary tract infections [9]. Breast cancer patients report the same genital symptoms, with the aggravating factor that their oncological condition worsens menopausal complaints or causes them to appear at a younger age, with enormous consequences on quality of life [4].

The pathogenesis of these disorders is based on the decrease in estrogen levels caused by the onset of menopause and by the adjuvant therapies among the breast cancer population. Hypoestrogenism leads to the thinning of the vaginal epithelium, a decrease in fibroblastic activity and collagen production, lower glycogen, and, consequently, an increased vaginal pH and impaired microbiome [10].

VVA is a chronic progressive condition requiring a long-term treatment, on cessation of which the symptoms tend to reappear [11].

For women with GSM, treatment can be approached gradually depending on the severity of the symptoms. First-line therapies for milder symptoms include vulvar and vaginal lubricants and moisturizers [12]. Low-dose vaginal estrogen therapy is the preferred pharmacological choice if symptomatic patients are not responsive to nonprescription therapies [13].

For breast cancer survivors, there are limitations in managing symptoms, and the conventional treatments include vaginal moisturizers and lubricants that provide temporary relief [14].

Physical methods for the treatment of VVA, such as vaginal laser therapy, represent a non-pharmacological second line option, particularly useful for women who are nonresponsive and/or noncompliant and those who have contraindications to hormones. All this considered, the use of laser technologies for these patients has been gaining ground during the last few years [15].

The primary purpose of this review is to narratively report the evidence regarding the efficacy and safety of Er:YAG and CO_2_ vaginal laser treatment for genital symptoms in women suffering from VVA.

### 1.1. Pathogenesis of Vulvovaginal Atrophy and Anatomopathological Changes after Menopause

The typical symptomatology of VVA is caused by the susceptibility of genital tissues to the decrease in estrogen levels. The genital and lower urinary tract share a common embryologic origin in women and widely express estrogen receptors [12]. In particular, vulvovaginal tissue normally presents alpha- and beta-estrogen receptors, but the latter are demonstrated to disappear after menopausal onset [16]. 

In the female genitals, the action of estrogen includes maintaining the thickness of the vaginal epithelium, the trophism of the smooth muscle layer, and the morphology and density of blood vessels and nerve endings. Within the extracellular matrix, there are fibroblasts responsible for producing collagen, which are also modulated by the action of estrogen. This mechanism is particularly important because collagen and other substances, such as proteoglycan macromolecules, provide elasticity and strength to the tissues [17]. 

With the advent of hypoestrogenism, the delicate nervous, muscular, and vascular mechanisms that regulate sexual function are disturbed. Estrogens, in fact, are involved in genital lubrication and trophism but also in complex brain networks that regulate sexual desire and satisfaction [18]. The microstructural changes caused by the estrogen withdrawal are accompanied by major anatomical and functional alterations: the epithelium becomes pale and less elastic, the vagina can narrow and shorten, the labia minora regress, and the introitus may constrict, all leading to severe sexual dysfunction [9].

Estrogens are also responsible for the maintenance of a physiological vaginal microbiome and a correct pH value. A healthy vaginal flora is characterized by a predominance of Lactobacillus species, which metabolizes glucose into lactic acid and acetic acid, lowering the vaginal pH to a range of 3.5–4.5 and protecting from vaginal and lower urinary tract infections [19]. With the thinning of the vaginal epithelium due to menopause, fewer squamous cells are discharged into vaginal secretions, and those that are have lower glycogen content. As vaginal glycogen levels fall, the population of Lactobacilli decreases and the vaginal pH increases [20]. 

### 1.2. Vulvovaginal Atrophy in Breast Cancer Survivors

Breast cancer is the most common neoplasia in women, accounting for 30% of all female malignancies, and invasive breast cancer has a lifetime probability of affecting one in eight women [20]. The increase in survival rate, which now exceeds 90% at 5 years [21], and the improvement in screening methods have led to the widespread diffusion of issues related to the quality-of-life impairments of these patients [22]. Menopausal symptoms affect up to 70% of breast cancer survivors [23] and mainly include vasomotor disorders and disturbances referable to atrophic vaginitis.

A major issue of this situation is the frequent young age of these patients, who often deal with premature menopause following adjuvant treatments for breast cancer. These disturbances can be particularly burdensome for such young women and can adversely affect their health, social, and intimate life [24].

The currently available therapies for breast cancer have increased the survival rates but have also caused a wide range of biological changes that result in medically induced menopause and, consequently, in quality of life impairment [25]. Up to 80% of breast cancers are estrogen-receptor-positive [26]: this characteristic has allowed the development of targeted therapies that have achieved satisfactory treatment results, such as aromatase inhibitors and tamoxifen. The former block the activity of the aromatase enzyme, which is responsible for converting androgens into estrogens, while the latter is a selective estrogen receptor inhibitor [27]. 

As a result of this pharmacological hormonal decline, both drugs are responsible for inducing menopausal symptoms, including significant vaginal dryness, dyspareunia, and subsequent sexual dysfunction [25]. Tamoxifen acts as an antagonist of estrogen-positive breast cells, although it has a partial agonist effect on estrogen receptor alpha in the vagina, causing a quasi-estrogenic consequence on genital tissue. This mechanism of action possibly explains why tamoxifen causes a lesser incidence rate of vaginal dryness compared to aromatase inhibitors [27].

### 1.3. Diagnosis and Assessment Tools for Vulvovaginal Atrophy

VVA is diagnosed via clinical evaluation and validated questionnaires. Considering the currently available literature, the most common assessment tools for GSM diagnosis are the visual analog scale (VAS) of VVA symptoms, the vaginal health index (VHI), and the female sexual function index (FSFI) [28].

While the VAS score is a progressive 10-point measure through which patients are asked to record their disturbances, the FSFI analyzes dyspareunia and general sexual disturbances with a six-domain structure form that includes questions about sexual desire, subjective arousal, lubrication, orgasm, satisfaction, and pain [29]. They both consider subjective outcomes and are influenced by patient perception of the disease. 

On the contrary, the VHI tends to be used as an objective measure, even if some authors disagree with this choice [28]. The VHI is a score that incorporates the evaluation of five elements: vaginal elasticity, secretions, pH, integrity of the epithelial mucosa, and tissue hydration. The score can vary between 5 and 25, with a cut-off of 15, below which is considered atrophic vaginitis [30]. Considering that four out of the five variables are, in fact, influenced by practitioners’ judgment, it is easy to understand why some authors refuse to use this tool as objective [28]. 

Vaginal pH measurement, which is part of the VHI but also a useful assessment tool itself, is calculated through a pH indicator strip [30]. Taking into account the mechanism of post-menopausal pH changes mentioned before, studies consider a value greater than 5.0 to be associated with decreased serum estrogens and menopause, with a correlation that has been compared even to that with an FSH decrease [31].

Another tool used for VVA diagnosis, even if less successful in the literature than VHI, is the Vaginal Maturation Index (VMI). It consists of a cytological examination and subsequent calculation of the percentage of parabasal, intermediate, and superficial squamous cells present on a vaginal smear [32]. The ratio of these cells is used both as a diagnostic tool and as an assessment for any possible positive change induced by therapy [30]. Lower scores are associated with greater estrogen deficiency and with a predominance of parabasal cells on the vaginal smear [32]. 

All the above-mentioned instruments are useful in the diagnostic process of VVA, but, as is evident, there is still no consensus on the definition of vaginal atrophy and, consequently, a unique diagnostic method.

Subjective outcomes are more useful in the clinical context as they are quick and immediately helpful for the subsequent therapeutic decision. Objective instruments should be applied in research settings and combined with symptoms evaluation [30].

The differential diagnosis of VVA must be made with all the conditions that imitate the symptoms and clinical appearance, such as lichen sclerosus, lichen planus, and vulvar malignancies [33].

### 1.4. Treatment Options Available for Vulvovaginal Atrophy

VVA is a chronic and progressive condition that does not resolve spontaneously and often worsens without treatment or on its suspension due to its pathogenesis and correlation with age progression and menopause [2]. The principles underlying the treatment of vulvovaginal atrophy are the recovery of physiological urogenital function and the improvement of symptoms.

Lifestyle changes are important as they act on risk factors that may accelerate estrogen deprivation and aggravate symptoms. For this reason, patients are advised to give up smoking, which reduces estrogen bioavailability, and to lose weight in the case of obesity, which is a condition that appears to decrease blood flow in the genitourinary area [34].

First-line therapies to relieve symptoms of GSM include over-the-counter non-estrogenic vaginal lubricants and moisturizers [12]. Lubricants may be water-, silicone-, or oil-based and are applied to external genitalia before sexual intercourse, providing relief from sexual discomfort [13]. Vaginal moisturizers offer greater genital hydration, and, according to a randomized controlled clinical trial, hyaluronic acid vaginal gel effectively improves clinical disturbances and may be considered as a valid alternative to estrogen-based treatments in relieving the symptoms of vaginal dryness [35]. However, although lubricants are able to alleviate symptoms, the improvement is often temporary, and frequent reapplications are necessary [36]. It should be noted that lubricants and moisturizers do not reverse urogenital ageing but compensate for its anatomo-functional consequences by improving sexual comfort and maintaining vaginal secretions [37]. 

While systemic estrogens are exclusively recommended for women who complain not only of genital symptoms but also of vasomotor disturbances and problems related to osteoporosis [12], low-dose vaginal estrogens are considered the gold standard for patients of vaginal atrophy and sexual dysfunction who are unresponsive to non-prescription therapies [38]. Vaginal dryness and dyspareunia are, in fact, the most common indications of low-dose local estrogen therapy [12]. With both creams and vaginal ovules, the therapeutic indication is one application per day for 2 weeks, followed by a maintenance dose of two to three applications per week. Ideally, women should be treated with the lowest dose and frequency at which symptom control can be achieved [39]. Despite the availability of a wide range of vaginal hormone products, a recent Cochrane review suggested that there is no conclusive evidence of a difference in efficacy between different preparations when compared to one another, but, more importantly, there is poor quality of evidence regarding the clinical efficacy when compared against placebos [40]. Other issues concerning vaginal estrogens are patients’ mistrust of hormone treatment and the low compliance with a daily-application therapy, which often leads women to abandon this prescribed medication [5].

An enormous disadvantage of both moisturizers and local estrogen therapy is precisely this poor compliance. A 2013 survey of more than 3000 post-menopausal women reported on patients’ experiences and perceptions of the available treatments: regarding all the vaginal administrated therapies, the women reported high dissatisfaction due to annoying application procedures and bothersome vaginal discharge. In addition to this, topical estrogen treatment was found to be burdened by concerns regarding long-term safety due to hormonal exposure and consequent oncological risk [7].

A hormonal alternative to vaginal estrogens is dehydroepiandrosterone (DEHA), whose vaginal insert was recently approved by the FDA for the treatment of GSM [13]. It is an intermediate steroid hormone in the biosynthesis of androgens and estrogens and has been demonstrated to be effective in improving VVA symptoms and vaginal pH without causing dangerous endometrial stimulation [41]. 

The only orally available product approved for the treatment of vaginal dryness and moderate to severe dyspareunia is Ospemifene, a selective estrogen receptor modulator with an agonist/antagonist effect [42]. A long-term efficacy and safety clinical study with 180 women showed sustained improvements regarding the symptoms and clinical examination of the vagina, with no cases of endometrial hyperplasia or malignancies [43]. There is still no full clarity on the possible side effects of Ospemifene as it has been shown to possibly cause a worsening of hot flashes and an increase in the risk of venous thromboembolism [44].

The main prescription therapies available for VVA with their pharmaceutical forms and active ingredients are summarized in Table 1.

For breast cancer survivors, the guidelines from the American Society of Clinical Oncology (ASCO)/American Cancer Society (ACS) [45] and the North American Menopause Society [38] recommend the use of nonhormonal therapies, such as lubricants and vaginal moisturizers, as first-line therapy for these patients. While systemic hormonal therapy in breast cancer survivors is contraindicated by international guidelines due to the lack of safety data [46], the use of vaginal estrogens is usually not recommended due to the possibility of their systemic absorption and consequent increase in hormonal blood levels; this effect could revert the hormonal suppression achieved by therapy and potentially stimulate occult breast cancer cells [47]. The safety of intravaginal DHEA and oral Ospemifene after breast cancer has not been fully established considering the dearth of long-term clinical investigations [24].

Considering the limitations of the current treatments for VVA, it is essential to provide an alternative for all those women who do not respond, have contraindications, or are not compliant with the previously mentioned available therapies.

### 1.5. Laser Functioning and Rationale for Treatment of Vulvovaginal Atrophy

The two main types of lasers currently used for the treatment of VVA are the fractional micro-ablative CO_2_ laser and the non-ablative photothermal Erbium:YAG laser.

Several other medical specialties started to use these technologies, with regenerative and rejuvenating purposes [48,49,50].

The effectiveness of the Er:YAG laser for vaginal atrophy was first described in 2015 [51]. Vizintin and colleagues reported this non-surgical, non-ablative thermal technique, which produces vaginal collagen hyperthermia following the remodeling and synthesis of new collagen fibers. This results in enhanced vaginal tissue tightness and elasticity and, consequently, improved symptoms of vaginal atrophy [51].

The vaginal micro-ablative CO_2_ laser was introduced in 2014 and, immediately, different histological studies [52,53] confirmed its efficacy in changing and rejuvenating vulvovaginal tissue in patients affected by VVA. Subsequent studies correlated this genital remodeling to vaginal atrophy symptoms improvements.

There are currently a growing number of studies on the effectiveness and safety of these two laser technologies in treating vaginal atrophy [22,54,55,56,57,58,59,60,61,62,63,64,65,66,67,68,69,70,71,72,73,74,75,76,77,78,79,80,81,82,83,84,85,86,87,88,89,90,91,92,93,94,95,96,97,98,99,100,101,102,103,104,105,106,107,108,109,110,111,112,113,114]. Although they are both employed for the same regenerative purpose, they are different in their mechanism of action and interaction with tissues.

Erbium:YAG technology is based on the concept of the controlled heating of the vaginal tissue, in particular the deeper mucosa, without over-heating the surface. A calibrated temperature stimulates the collagen fibers to contract and, consequently, the surrounding tissue to shrink as well. The thermal effect then continues throughout the processes of collagen remodeling and neocollagenesis, resulting in the generation of new fibers and an overall improvement in the tightness and elasticity of the treated tissue [51].

The micro-ablative fractional CO_2_ laser exploits the heat generated by the vaporization of water in the cells in the deeper lamina propria [60]. Energy and, consequently, the micro-ablative impact are precisely delivered in order to limit surrounding tissue damage. The ultimate effect of this hyper-regulated injury includes neocollagenesis and neovascularization, with consequent improvements in vaginal pH, moisture, blood flow, and ground substance turgidity [76].

The mechanism of function of these two laser technologies and the tissue changes produced by the action of both have led to a realization of their potential for treating vulvovaginal symptoms caused by hypoestrogenism.

New collagen formation, the restoration of its architecture, neovascularization, and production of a ground matrix can contribute to reducing vaginal laxity while restoring the hydration to a more physiological vaginal pH and recreating a protective film that constitutes a barrier to genital infection. All these tissue changes represent a real rejuvenating process of the vaginal wall that is also demonstrated at the ultrastructural level [52].

### 1.6. Laser Treatment Results in Patients Affected by VVA

The available literature, so far, counts several clinical studies on the efficacy and safety of the vaginal laser for the treatment of VVA, most of them concerning the use of the micro-ablative fractional CO_2_ laser [22,54,55,56,57,58,59,60,61,62,63,64,65,66,67,68,69,70,71,72,73,74,75,76,77,78,79,80,81,82,83,84,85,86,87,88,89,90,91,92,93,94,95,96,97,98,99,100,101] and others focusing on Erbium:YAG vaginal laser treatment [103,104,105,106,107,108,109,110,111,112,113,114].

Most of those selected are non-randomized prospective studies [52,54,59,60,61,62,64,66,67,68,69,70,71,73,74,78,79,82,83,86,90,96,98,100,102,103,104,108,110,114]. A smaller proportion are randomized clinical trials [56,57,91,92,93,94,95], and others are observational retrospective [58,61,63,65,77,80,81,84,88,89,97,99,113] and pilot studies [54,71,75,84,86,104,105,106,108,110,111].

All the studies deal with laser treatment for vulvovaginal atrophy in patients affected by VVA. Most of the treated patients were post-menopausal women suffering from GSM, but some studies are based on treatment for breast cancer survivors [22,81,82,83,84,85,86,87,88,89,98,100,103,104,109] suffering from genital disorders due to atrophy.

The standard treatment protocol includes three laser treatments 30–40 days apart, although one study analyzed the possible beneficial effect of additional fourth and fifth sessions [74].

The symptoms significantly improve in almost all the investigations, with a significant reduction in the VAS scores for vaginal dryness, burning, itching, and dyspareunia and an augmentation in the total FSFI and FSDS scores, but also in their individual items. Several investigations evaluate the vaginal CO_2_ laser long-term efficacy and safety and demonstrate maintained improvements at 12, [59,60,70,79,87,88,96,97], 18 [100], and 24 months follow-up [98]. Isaza and colleagues even demonstrate that improvements are maintained for up to 36 months without the need for any further intervention [102]. For Er:YAG, fewer studies reach these long-term evaluations, and they show that the values of subjective and objective assessments return to a level similar to the baseline after 18 to 24 months [107].

When taken into account, the VHI further confirms the effectiveness of laser treatment, increasing its value significantly and adding an objective assessment to the subjective improvement of patients after therapy. Moreover, the pH value differs significantly after treatment, becoming more acidic and returning to levels similar to those in pre-menopause [100,103,110,112,113]. When authors decide to analyze the effect of laser on cell maturation through the VMI score, they find significant improvements in both cytology and symptoms relief [61,64,74,75,93,94,95,112]. Only Takacs et al., investigating vaginal cytology after laser treatment, report that the VMI score does not change significantly, but, despite this, all those patients who demonstrate an improvement in this score have significantly reduced symptoms after therapy [90].

Interestingly, all the studies comparing a laser with a sham show an improvement in the symptoms even in the sham-arm, suggesting a possible placebo contribution [54,55,56].

No serious adverse event was found in any of the evaluated studies. The most common reactions are a transitory burning sensation [61,68,105,109,112,113] and slight pain during the procedure [55,56,68,84], mostly due to the probe insertion and rotation. The procedure-related discomfort is demonstrated to be significantly lower after the three cycles of laser treatment, and the tolerability increases with the genital atrophy improvement [81]. These considerations prove that one of the most important advantages of vaginal laser treatment is the possibility of its execution on an out-patient basis, without even the need for local anesthesia.

On 30 July 2018, the United States Food and Drug Administration (FDA) issued a claim against the use of energy-based devices, such as lasers and radiofrequency, to perform vaginal rejuvenation, vaginal cosmetic procedures, and treatments for genital disorders related to menopause or sexual dysfunction [115]. For this reason, this newly established safety of laser treatment is a key element to be emphasized.

## 2. Conclusions

Vulvovaginal atrophy is a major problem in gynecological practice due to its very widespread occurrence and the major negative impact it has on the quality of life of the women affected. Considering that it is a chronic and progressive disease, it is easy to see how finding valid therapeutic alternatives for these patients is of fundamental importance. Given the current limitations of the available therapies in terms of efficacy, safety, and compliance, it is essential to find innovative tools that can be useful in this field.

Vaginal laser treatment represents a valid, innovative, and minimally invasive therapy for the treatment of vulvovaginal atrophy symptoms as it has been widely demonstrated to be effective and completely safe.

In almost all the studies available in the literature, there is a statistically significant improvement in all the vulvovaginal symptoms and in the patients’ sexual function. This observation is made immediately after treatment and also tends to remain valid upon long-term follow-ups.

This up-to-date literature review thus confirms the effectiveness of a vaginal laser not only for women with genital atrophy related to menopause but also in breast cancer survivors, for whom several restrictions in treatment possibilities make management more difficult.

No study at this stage has been able to detect predictive characteristics of the response to laser therapy. This may be important for a more individualized therapy for vulvovaginal atrophy and possibly intervening in modifiable factors to improve the laser treatment response.

## Figures and Tables

**Table 1 medicina-58-00770-t001:** Main pharmaceutical options for VVA treatment.

	Administration Route	Formulation	Active Ingredients
**Systemic estrogens**	Oral	Tablets	Estradiol
	Transdermal	Patches	Conjugated estrogens
**Vaginal estrogens**	Topical	Vaginal ovules	Estradiol
		Vaginal cream	Estriol
		Vaginal ring	Conjugated estrogens
**Ospemifene**	Oral	Tablets	Ospemifene
**DEHA**	Topical	Vaginal ovules	Prasterone

## Data Availability

Data sharing is not applicable to this article.

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
