# Peer review of "New Innovations for the Treatment of Vulvovaginal Atrophy: An Up-to-Date Review"

_medicina, 2022, doi:10.3390/medicina58060770_

Round 1

Reviewer 1 Report

The topic presented is current and essential to women's health. 

The authors provide an adequate theoretical framework for the topic of vulvovaginal atrophy.

Regarding the description of symptoms, the authors list some of the most common symptoms of vulvovaginal atrophy. However, the review would benefit from citing other works that support the relevance of these symptoms expressed in the percentages found in the different studies. 

The reference to women suffering from this clinical condition due to breast cancer intervention is not entirely in line with the rest of the introduction. However, it must be mentioned in the document, and it would benefit from being integrated with the rest of the introduction. 

The authors clearly indicate the objective of the review work.

The approach to atrophic vaginitis in menopausal women and women who have had breast cancer is evident and objective. The authors correctly highlight the most important physiological processes of each of the conditions. 

The topic "Treatment options available for vulvovaginal atrophy" is not clear what the authors mean by "VVA is a chronic condition that often progresses without treatment or on its suspension" (lines 179-180). Authors should clarify.

Although the topic regarding the treatment used to manage vulvovaginal atrophy is clear and generically addresses all the therapeutic strategies available on the market, the work would benefit from a summary table integrating their pharmaceutical forms and active ingredients. In addition, the acceptability of these types of products could be discussed more extensively. 

The last topic, "Laser treatment results in patients affected from VVA", is supported by the literature with an extensive literature review. This topic clearly differs from the others. It would have been important for the authors to use the same strategy for the rest of the document. 

Some of the topics discussed, except "Laser treatment results in patients affected from VVA", show a discussion and enumeration of essential points in the topic but lack a correct literature citation. The authors, in most cases, have limited themselves, for example, to referring to and citing other reviews. Therefore, it will be necessary to indicate the references to the original works.

Author Response

First of all, thank you so much for all your comments. We have tried our best to revise and clarify all the points you have highlighted to us, with the aim of improving the work done thanks to your contribution.

For this reason, we have added a wider range of symptoms prevalence by drawing on other studies concerning VVA epidemiology and we have extended the description concerning the issue of breast cancer patients to ensure a better presentation of the problem in the introduction.

We later clarified the sentence "VVA is a chronic condition that often progresses without treatment or on its suspension" (lines 179-180)", making it clear that the fundamental problem with this disease is that, due to its pathogenesis, it does not resolve on its own but requires chronic treatment, upon discontinuation of which symptoms tend to reappear.

We provided a table including the main available pharmacological treatment options with their pharmaceutical forms and active ingredients. We chose to make it very simple, immediate and summarizing, so that it could be a way of schematically visualizing the alternatives, although this is not the main theme of our work. In addition to that, as suggested, we discussed more extensively their acceptability (in particular for the topical administered options), considering that this is one of the most important issue concerning patients' adhesion to VVA treatment in general.

Regarding the difference noted in the presentation of the available literature on the vaginal laser treatment compared to that on other possible therapies, we acknowledge that in this manuscript the  references quoted for lasers are much more compared to classical options. We did it on purpose, since, exactly as you recognized and appreciated, the objective of the review work is about innovative therapy, and specifically as mentioned in the introduction, on laser treatment. Moreover we do believe that it is important to emphasize the increasing number of papers on this issue since, in the past, laser has been criticized for limited evidences.

Reviewer 2 Report

Thanks for submitting this work. There are still alot of confusion about laser effectiveness in GUS. The effect of estrogen is well described in the text. Concerning the maturation effect of estrogen on vaginal mucosal cells, how come with laser obtaining the same results with estrogen by means PH values.Because in the text it was mentioned'' VMI did not show significant change'' with laser therapy. Without change in VMI ,it is hard to achieve maturation and PH normalization.This paragraph needs some discussion.

Author Response

First of all, thank you so much for all your comments. We have tried our best to revise and clarify all the points you have highlighted to us, with the aim of improving the work done thanks to your contribution.

We regret that we did not immediately make clear the issue of changes in the VMI score after laser treatment. However, thanks to your comment, we have now explained the evidence regarding cytological changes more extensively, specifying that almost all the papers that have analyzed changes in the VMI score have found a significant improvement after laser treatment and that even the only paper citing an absence of a statistically significant difference reports that the patients with cytological improvement are those with better symptom improvement.

This manuscript is a resubmission of an earlier submission. The following is a list of the peer review reports and author responses from that submission.